# Causal Attribution of Model Performance Gaps in Medical Imaging Under Distribution Shifts

## Abstract

Deep learning models for medical image segmentation suffer significant performance drops due to distribution shifts, but the causal mechanisms behind these drops remain poorly understood. We extend causal attribution frameworks to high-dimensional segmentation tasks, quantifying how acquisition protocols and annotation variability independently contribute to performance degradation. We model the data-generating process through a causal graph and employ Shapley values to fairly attribute performance changes to individual mechanisms. Our framework addresses unique challenges in medical imaging: high-dimensional outputs, limited samples, and complex mechanism interactions. Validation on multiple sclerosis (MS) lesion segmentation across 4 centers and 7 annotators reveals context-dependent failure modes: annotation protocol shifts dominate when crossing annotators ($7.4\% \pm 8.9\%$ DSC attribution), while acquisition shifts dominate when crossing imaging centers ($6.5\% \pm 9.1\%$). This mechanism-specific quantification enables practitioners to prioritize targeted interventions based on deployment context. Our code is available at anonymous repository.

## 1 Introduction

Medical image segmentation models excel in controlled settings but exhibit unpredictable performance drops in clinical deployments [1, 2]. Unlike classification tasks where shifts have been studied [3, 4, 5], segmentation presents unique challenges: spatial correlations, high-dimensional outputs that interact non-linearly with distribution shifts, etc.[1, 6]. Consider a white matter lesion (WML) segmentation model underperforming at a new hospital. The failure could stem from scanner changes (acquisition shift), inconsistent radiologist annotations (annotation shift), or demographic changes (population shift) [1]. Existing domain generalization methods treat these shifts monolithically, offering no insight into which mechanisms drive performance degradation [7]. We address this gap by extending causal attribution frameworks [3, 5] from low-dimensional classification to high-dimensional segmentation tasks. Our approach leverages the principle of Independent Causal Mechanisms (ICM) [8] to model the medical imaging data-generating process (DGP) [1, 9], and employs Shapley values to quantify each mechanism's contribution to performance drops.

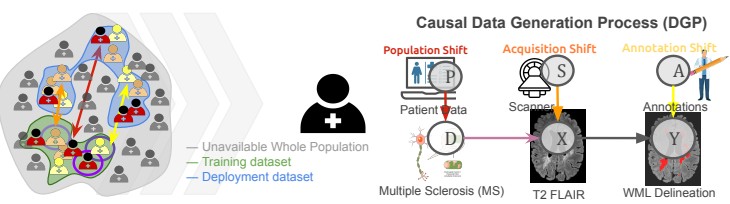

Figure 1: Causal modeling of domain shifts in medical imaging. We attribute performance degradation to shifts in acquisition ($P(X|S, D = \mathrm{MS})$) versus annotation ($P(Y|X, A)$) mechanisms.

## 2 Methods

We model the DGP for segmentation task via a causal graph as in Figure 1. Following the ICM principle [8, 10], the joint distribution factorizes as: $P(V) = \prod_{i=1}^{n} P(V_i \mid \mathbf{PA}_i)$, where $V = \{V_1, \ldots, V_n\}$ represents the system variables (demographics, images, annotations), and $\mathbf{PA}_i$ denotes the parent variables of $V_i$. This factorization remains structurally invariant across environments, though the individual mechanism, $P_{V_i|PA_i}$, distributions may shift. Then, let $f$ denote a model trained on data from a training environment $\epsilon_{tr}$, later deployed in environment $\epsilon_{dep}$, and $M$ an assessment metric. Performance change between $\epsilon_{tr}$-$\epsilon_{dep}$ is defined as $\Delta M = M(f, P^{\epsilon_{tr}}) - M(f, P^{\epsilon_{dep}})$. $\Delta M$ can be causally attributed to shifts in the distributions of individual mechanisms. Through a causal lens, the transition from $\epsilon_{tr}$ to $\epsilon_{dep}$ is explained through a set of intervened (shifted) mechanisms. For any subset of mechanism indices $\mathcal{I} \subseteq \{1, 2, ..., n\}$, we define a mixed distribution $P_{\mathcal{I}}$ where only mechanisms in $\mathcal{I}$ are intervened upon:

$$P_{\mathcal{I}}(V) = P(V_{i \notin \mathcal{I}}, \text{do}(V_{i \in \mathcal{I}} = V_i^{\epsilon_{dep}}))) = \prod_{i \in \mathcal{I}} P_{V_i|PA_i}^{\epsilon_{dep}} \prod_{i \notin \mathcal{I}} P_{V_i|PA_i}^{\epsilon_{tr}} \tag{1}$$

This represents the distribution that would result if we selectively transported only the mechanisms indexed by $\mathcal{I}$ from the deployment environment while keeping all other mechanisms at their training state, which will cause an estimated change of $\Delta M_{\mathcal{I}} = M(f, P^{\epsilon_{tr}}) - M(f, P_{\mathcal{I}})$,

**Shapley Symmetry.** This formulation allows us to systematically decompose $\Delta M$ into contributions from individual mechanisms. However, the contribution of each mechanism to performance drop depends on the order in which mechanisms are shifted. For instance, altering annotation protocols before scanner parameters may yield different marginal impacts than the reverse sequence. This path-dependence, where mechanism shifts propagate non-additively, requires fair attribution. To ensure it, we employ Shapley values [11, 5] to symmetrize over all possible intervention sequences:

$$\phi_i(\Delta M) = \sum_{\mathcal{I} \subseteq \{1,2,...,n\} \setminus \{i\}} \frac{|\mathcal{I}|!(n - |\mathcal{I}| - 1)!}{n!} \left[ \Delta M_{\mathcal{I} \cup \{i\}} - \Delta M_{\mathcal{I}} \right] \tag{2}$$

$\Delta M_{\mathcal{I}}$ **Estimation.** A fundamental challenge is that the distributions $P_{\mathcal{I}}$ are not directly accessible; we only have samples from $P^{\epsilon_{tr}}$ and $P^{\epsilon_{dep}}$. Computing $\Delta M_{\mathcal{I}}$ requires evaluating model performance under counterfactual mechanism combinations that take combinatorial complexity. To address this, we use importance sampling to reweight samples from the training distribution,

$$M(f, P_{\mathcal{I}}) = \mathbb{E}_{(V_{\mathcal{I}}) \sim P_{\mathcal{I}}}[M(f, P_{\mathcal{I}})] \approx \mathbb{E}_{(V_{\mathcal{I}}) \sim P^{\epsilon_{tr}}}[w_{\mathcal{I}} M(f, P_{\mathcal{I}}))],$$

where $w_{\mathcal{I}}(x, y)$ represents importance weights, $w_{\mathcal{I}}(x, y) = \frac{P_{\mathcal{I}}(x,y)}{P^{\epsilon_{tr}}(x,y)} = \prod_{i \in \mathcal{I}} \frac{P^{\epsilon_{dep}}(V_i|\mathbf{PA}_i)}{P^{\epsilon_{tr}}(V_i|\mathbf{PA}_i)}$.

In medical image segmentation, this allows us to estimate how performance would change if, for example, only the annotation protocol shifted while scanner parameters remained constant. For example, when evaluating WML segmentation across hospitals, we can isolate the effect of annotation style differences by constructing weights that capture only the shift in $P(Y|X, A)$ (annotation mechanism) while keeping $P(X|S)$ (image acquisition mechanism) fixed. To estimate these importance weights, we train binary classifiers to discriminate between environments for each mechanism following [12]. For mechanism $i$, we train a classifier $\mathbf{D_i}$ to predict whether a sample comes from $\epsilon_{tr}$ or $\epsilon_{dep}$ based on $(V_i, \mathbf{PA}_i)$. The density ratio can then be expressed as, $\frac{P(\epsilon_{dep}|V_i, \mathbf{PA}_i)}{P(\epsilon_{tr}|V_i, \mathbf{PA}_i)} \cdot \frac{P(\epsilon_{tr})}{P(\epsilon_{dep})}$

**Discriminator Training, $\mathbf{D_i}$.** Training robust discriminators $D_i$ for shift detection presents unique challenges in medical imaging contexts. To mitigate overfitting, we implement gradient penalty regularization [13] and employ a multi-scale architectural design that captures both local and global distribution shifts. Additionally, we utilize test-time augmentation during discriminator training to enhance stability when handling the limited sample sizes common in medical datasets. Our implementation is fully integrated within the *nnU-Net* framework [14].

## 3 Experiments and Results

**Experimental procedure:** We train *nnU-Net* segmentation models on source data ($\epsilon_{tr}$) and test on target ($\epsilon_{dep}$), measuring $\Delta M$ using Dice Similarity Coefficient (DSC) and F1 score. Discriminators

72 $D_i$ estimate density ratios enabling importance sampling to compute counterfactual performance
73 under selective mechanism shifts, aggregated via Shapley values into per-mechanism attributions. We
74 evaluated on MSSEG2016 [15, 16], comprising 53 MS patients from 4 centers with 7 annotators, with
75 documented inter-rater variability and scanner heterogeneity. We designed two experiments: **Exp. A**
76 trains on annotator $i$ and tests on annotators $j \neq i$ (annotation shifts), while **Exp. B** trains on centers
1,7,8 and tests on center 3 (acquisition shifts). Table 1 shows distinct mechanism contributions across

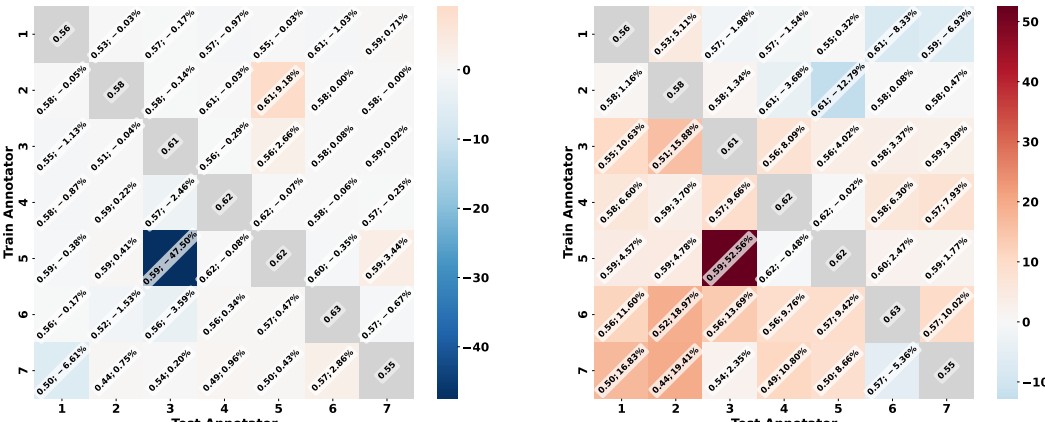

Figure 2: Inter-annotator performance for Exp. A. Each cell shows DSC; $\Delta_{DSC}$. (a) Acquisition
mechanism $P(X|S)$ shows predominantly negative $\Delta_{DSC}$, indicating minimal or positive contribution. (b) Annotation mechanism $P(Y|X,A)$ exhibits predominantly positive $\Delta_{DSC}$.

77
78 environments. In Exp. A (annotator shifts), the annotation mechanism $P(Y|X,A)$ contributes $7.4\% \pm$
79 $8.9\%$ (DSC) and $12.8\% \pm 14.8\%$ (F1) to performance changes, while the acquisition mechanism
80 $P(X|S)$ shows $1.6\% \pm 7.1\%$ and $5.8\% \pm 12.3\%$. Negative $\Delta_{DSC}$ values in the $P(X|S)$ mechanism
81 indicate performance improvements rather than degradation. In Exp. B (image shifts), the relative
82 contributions reverse: acquisition mechanism $P(X|S)$ contributes $6.5\% \pm 9.1\%$ (DSC) and $14.2\% \pm$
83 $12.9\%$ (F1), while annotation mechanism shows $2.6\% \pm 5.8\%$ and $8.4\% \pm 9.8\%$. Figure 2 visualizes
84 the full attribution matrix for Exp. A, revealing heterogeneous annotator sensitivity with $\Delta_{DSC}$
85 ranging from minimal values to $52.6\%$.

Table 1: Mechanism Contributions to Performance Changes (%)

| Exp. | Mechanism | $\Delta_{DSC}(\%)$ | $\Delta_{F1}(\%)$ | | Exp. | Mechanism | $\Delta_{DSC}(\%)$ | $\Delta_{F1}(\%)$ |
|------|-----------|--------------------|-------------------|---|------|-----------|--------------------|-------------------|
| A | $P(Y|X,A)$ | $7.4 \pm 8.9$ | $12.8 \pm 14.8$ | | B | $P(Y|X,A)$ | $2.6 \pm 5.8$ | $8.4 \pm 9.8$ |
| | $P(X|S)$ | $1.6 \pm 7.1$ | $5.8 \pm 12.3$ | | | $P(X|S)$ | $6.5 \pm 9.1$ | $14.2 \pm 12.9$ |

## 4 Discussion and Conclusion

87 We extend causal attribution to medical image segmentation, addressing its unique challenges. Our
88 findings reveal that dominant failure mechanisms depend critically on deployment context. In Exp.
89 A, annotation mechanism contributes 2-3 times more to performance changes. This pattern reverses
90 Exp. B, where acquisition shifts dominate. This has direct implications for resource allocation: when
91 deploying across institutions with different annotation protocols, prioritize annotation standardization;
92 when deploying to new scanner types, focus on scanner harmonization. While our experiments aimed
93 to isolate individual mechanisms, real medical datasets contain inherent confounding that cannot be
94 fully eliminated. In Exp. A, the acquisition mechanism still contributes $1.6 - 5.8\%$, likely because
95 different annotators labeled different case subsets or temporal annotation drift occurred. Importantly,
96 the shifted mechanism dominates (1.7-3 times higher attribution). This residual attribution reflects
97 real-world deployment where mechanisms rarely shift in complete isolation. Our approach requires a
98 known DGP, sufficient samples for discriminator training, and assumes static mechanisms, limiting
99 applicability. Future work should validate attribution accuracy using controlled synthetic experiments
100 where ground truth is known, enabling evidence-based deployment strategies.

## Potential negative societal impacts:

Over-reliance on attribution results without clinical context could lead to premature deployment decisions. The framework's requirement for deployment data may exclude resource-limited institutions, potentially widening healthcare disparities. Additionally, focusing solely on dominant mechanisms might overlook rare but critical failure modes affecting minority patient subgroups.

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
