# OpenReview forum: "Causal Attribution of Model Performance Gaps in Medical Imaging Under Distribution Shifts"
_EurIPS.cc/2025/Workshop/MedEurIPS — EurIPS 2025 Workshop MedEurIPS Submission_

### Official Review · Reviewer_kHCf · 2025-10-28
**Review comments.**

**Rating:** 6
**Confidence:** 4

**Review:**

In this paper, the author presents a causal lens of medical image segmentation data generation process and attribute the sharply value for evaluating performance changes attribute to individual mechanisms shifts. Experimental results reveals important interventions based on deployment context.

Minor issues:
1. Figure 1 (left) is visually chaotic. The meaning of the colored bi-directional arrows is unclear and should be explained or simplified for better comprehension.
2. i am more interested in future works that leverages their empirical findings to propose concrete methods for addressing distribution shifts in medical segmentation model training.

Overall this paper is clear and straightforward, and the proposed causal lens for distribution shift could be beneficial for medical AI community.

---

### Official Review · Reviewer_ENvT · 2025-10-29
**Review Comments**

**Rating:** 6
**Confidence:** 4

**Review:**

This paper extends causal attribution frameworks to medical image segmentation, using Shapley values to quantify how different mechanisms (acquisition protocols vs. annotation variability) independently contribute to performance degradation under distribution shifts.

Several aspects could be improved:

1. The work lacks ablation studies to justify the necessity of using Shapley values over simpler marginal attribution methods.
2. The paper would benefit from qualitative examples that illustrate the effectiveness of the proposed approach.
3. The causal definition should be described more clearly.  Like the rationale for why Multiple Sclerosis does not directly affect annotation? It would also help to clarify how the proposed causal graph was derived, whether it was inspired by expert domain knowledge, causal discovery methods...

---

### Decision · Program_Chairs · 2025-10-31

**Decision:**

Accept (Poster)

**Comment:**

Both reviewers find the paper clear and relevant, offering a valuable causal perspective on performance degradation under distribution shifts.